# Perspective of Local Government on the Performance Assessment of District Sports and Leisure Centers

**Chin-Yi Fred Fang**

The Graduate Institute of Sport, Leisure, and Hospitality Management, National Taiwan Normal University, Taipei City 106, Taiwan; fred@ntnu.edu.tw; Tel.: +886-2-77491448

**Abstract:** This paper addresses the problem of performance management of operate-transfer (OT) project finance for public sports and leisure centers (SLC) from the perspective of local government. This study contributes to use an evolutionary theory of competitive advantage and mixed-methods, including a modified Delphi method to develop the efficiency-based performance model (EPM) under OT project finance for the public SLCs. The total-factor framework disaggregating the efficiency into an innovative output surplus target ratio (OSTR) provides local governments with a contracted period to manage the SLCs through further specific improvement advice. This study further proposed the four-quadrant matrix formulated by long-term efficiency and short-term profitability to identify the benchmark and improvement directions. The empirical results indicated that there are fifteen SLCs located in the benchmark quadrant. This study provides policy makers in the local governments with a scientific reference to keep or drop the current operating private enterprise in the next concession period. The most underperforming SLCs could follow this proposed quadrant analysis and OSTR index, utilizing their internal resources to develop more attractive and reasonable-price exercise courses for participant growth.

**Keywords:** evolutionary theory of competitive advantage; efficiency-based performance model (EPM); quadrant-based matrix; data envelopment analysis; output surplus target ratio

## 1. Introduction

Most studies focus on the efficiency improvement of local services through reducing local public expenditure shared by many local governments around the world. Local governments could benefit from improving the efficiency and effectiveness of public services and simultaneously reducing public spending [1]. Meanwhile, local governments are under a lot of pressure from central governments to implement innovative actions to utilize resources more efficiently by trying unprecedented forms of management [2]. However, there is limited research to increase efficiency through maximizing social welfare and at the same time local government revenue.

Sports and exercising are able to yield many positive outcomes, including healthier communities, raising standards and economic vitality, and meeting the needs of children and young people [3]. Public sports and leisure facilities are a very crucial indicator of local government provisions, and they provide a major contribution to these beneficial outcomes as well as leading to the outcome quality [4]. Hence, how to benchmark success for sports and leisure centers (SLCs) is also one of the most important empirical topics from the commercial enterprise and government perspectives. Kung and Taylor [5] documented that over 60% of local authorities' leisure department net expenditures in the United Kingdom go to indoor sports facilities. They indicated that indoor sports facility investment is mandatory. Liu, Taylor, and Shibli [6] indicated that the local government provided public sports facilities, including swimming pools and leisure centers, at subsidized prices in the UK. However, Taiwan's civil sports and leisure centers (SLCs) were built

using central or local government funding and then were operated and transferred (OT) to private firms within the contracted years in order to spur their operating performance and generate revenue for the local authority. This is in contrast to the SLCs subsidized by local governments in the UK. Rossi, Breuer, and Feiler [7] defined those commercial sports providers as profit-oriented organizations. In contrast to commercial sports providers, non-profit sports clubs do not endeavor to generate profits but rather offer opportunities for active sports participation. Due to the OT management type, the SLCs in Taiwan have both the operating goals of revenue or profit generation and sports participant growth. This unique business OT environment is quite different from other countries and allows us to investigate their performance using this OT management style. There is a paucity of research to propose a new framework on a multi-factor performance assessment for the public sports and leisure facilities using OT project finance.

Service quality has been the focus of the extant studies on sports centers [4,8–11], while an investigation of operating efficiencies has been neglected. Due to the competitiveness of public SLCs and private fitness clubs, to date there is relatively little empirical evidence on the operational performance of SLCs. Especially, there is a limited framework for the local government's decision to keep or change the current private merchant in the next contracted period under the OT project finance. Even though Iversen [12] assessed the impact of strategic behavior of the private non-profit sports facility on utilization, to the best of the author's knowledge, this paper is the first limited attempt to build an assessment model of operating performance in civil SLCs for the local government and operating company, and it fills the literature gap about the performance evaluation of sports facilities.

This paper utilized an evolutionary theory of competitive advantage [13,14] to establish the input–output multi-factor performance assessment model. Donnellan and Rutledge [15] have documented that an organization has a bundle of resources and capabilities. Resources encompass tangible elements such as assets and equipment and intangible elements such as human resources [16]. An organization's capability is measured by its proficiency to organize inputs available to it in order to accomplish the preferred outputs [17] and its revenue-enhancing capacities [18]. The focus of performance assessment has moved from an estimation of performance in terms of simple indicators to the perspective of multidimensional systems [19]. While estimating an organization's performance, several studies [20,21] claimed that a multifactor performance assessment model should be developed, because 'performance' is a complex indicator requiring more than just any single measure to depict it. Hong and Jeon [22] further indicated that efficiency is equivalent to performance, and they used the data envelopment analysis (DEA) model to assess efficiency scores to reflect the business value creation process. As corporate sustainability is generally defined as a business approach to create long-term shareholder value with the limited resources, Lo [23] further presents that an organization's sustainability in terms of generating profit and attracting more investors has not been investigated a great deal. Business sustainability had been operationalized to create shared value for both business and society (e.g., Ortiz-de-Mandojana and Bansal [24]). Porter and Kramer [25] indicated that business managers pursuing long-term sustainability might have less risk than pursuing short-term financial performance. Barney [13] indicated that some of these efficient and effective routines in a firm operate sustained competitive advantage in the face of competition. This paper therefore adopted the evolutionary theory of competitive advantage to assess the performance of the routines with input investments to generate public and private financial outcomes (revenue) and social value (number of participants in the SLCs) to estimate the multi-dimensional sustainable efficiency. This type of multi-factor efficiency score could be valued as long-term sustainability [26]. Meanwhile, this paper used the short-term financial indicator, net income, as another axis to develop a quadrant-based matrix including sustainability and profitability of SLCs to identify and benchmark sustainable and profitable SLCs.

Honma and Hu [27] propose that DEA is the best tool for performance assessment and explicitly indicate the potential saving of inputs through efficiency calculation. Nevertheless, they mentioned that aggregate efficiency scores cannot determine which of these resources need improvement. Furthermore, more in-depth analysis requires disaggregated data for efficiency

assessment [27,28]. Therefore, Hu and Wang [28] developed the total factor framework-total-factor energy efficiency (TFEE) indicator as the ratio of the target energy input, as suggested by the DEA, to the actual energy input.

However, in terms of performance assessment of SLCs, these studies suggested that maximization of outputs or the output-oriented DEA model is more appropriate [5,29]. Hence, this paper develops the output surplus target ratio (OSTR), defined as the ratio of the target output, as also estimated by the DEA, to the actual output.

The contribution of this paper comes from using a multiple output–input framework in the form of the efficiency frontier to understand the conversion of resources into achieving both objectives fulfilling OT finance (one goal of revenue is for private business, another goal of number of participants is for government) in a service business. The aims herein are to develop the efficiency-based performance model of SLCs, empirically assess the performance of each SLC in Taiwan, and provide the benchmark and improvement direction for inefficient and unprofitable SLCs via establishing a four-quadrant analysis to differentiate the sustainable benchmark quadrant (high efficiency and high net income), "change operating" quadrant (low efficiency and low net income), and quadrants for improvement in efficiency and net income so as to improve the less well-performing SLCs. Furthermore, this study develops an innovative OSTR indicator to disaggregate the efficiency score into different output achievement targets for further specific improvement in order to aid inefficient SLCs to achieve the performance frontier.

This paper is structured as follows: after the introduction, this article sets out the institutional framework and literature review in support of the study. Section 3 demonstrates the theoretical background and literature review for performance assessment, and Section 4 articulates the qualitative and quantitative methods to measure the efficiency-based performance assessment model by applying DEA and to develop the quadrant-based matrix in order to identify the benchmark and improvement clusters. The total-factor framework to assess the OSTR for decision making is further provided. Section 5 illustrates the empirical result and matrix based on long-term efficiency and short-term profitability. Section 6 presents the discussion and an implication for SLCs. Section 7 concludes and indicates future research.

## 2. Public Administration on the District Local Sports and Leisure Centers in Taiwan

Local government used to promote participation and achievement in sport and leisure activities to support a healthy lifestyle for the citizens through physical activity (PA). Establishing the district SLCs near the residents is one of the effective ways to provide a range of services, initiatives, and programs to fully support resident increases to their PA in safe and inclusive environments. There are twelve districts in Taipei City in Taiwan in Figure 1. The local government in Taipei City established and managed these twelve SLCs in each district through an operate-transfer (OT) project, which involves a private partner participating in the operations of the public SLCs but without involving the task of SLC's construction [30]. The British local government provided public sports facilities at subsidized prices [6]. However, Taiwan's civil SLCs were built up through central or local government funding and then were operated and transferred to private firms within concession years in order to make their operating more efficient and generate higher revenue for the local authority, in contrast to the SLCs subsidized by the local government in the UK. The operate-transfer project finance model used public–private partnership as a business model to support government provision and deliver the services through private firms [31]. The aims of the Department of Sports under the Taipei City Government are to plan, construct, manage, and maintain public sports venues and facilities in Taipei City, to establish sets of authoritative regulations, to supervise the government outsourcing to private corporations to improve the profitability of sports industry, and to increase the operational efficiency and satisfaction of the sport facilities [30].

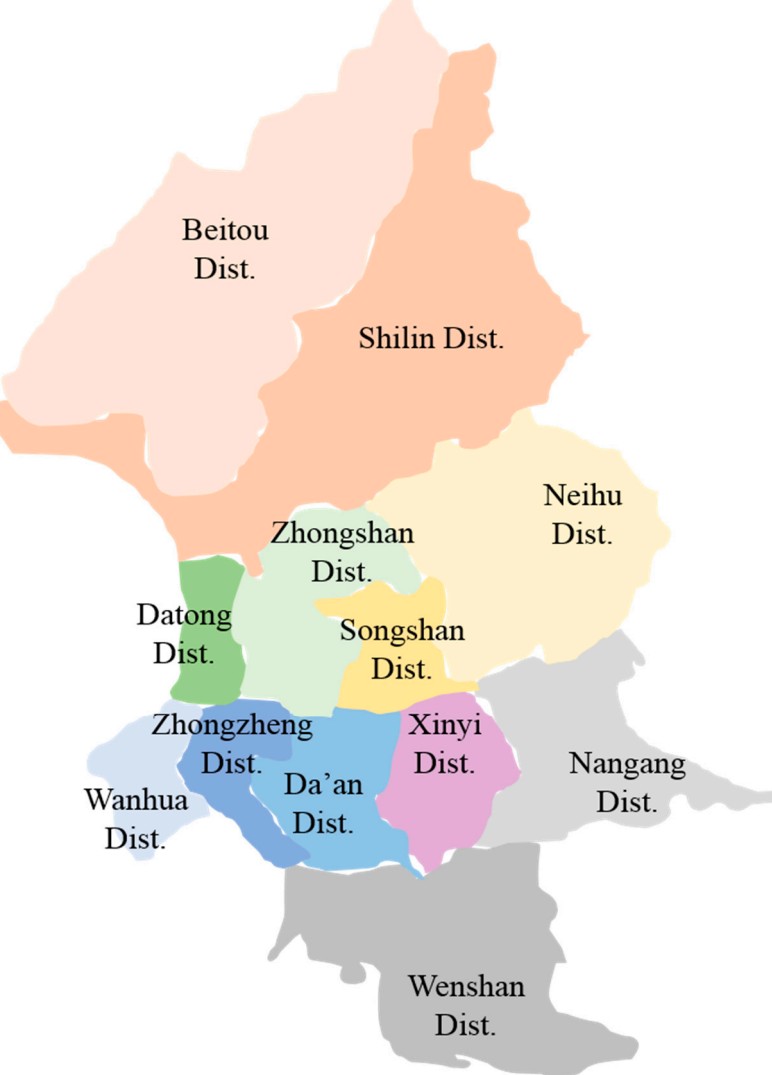

**Figure 1.** Twelve districts in Taipei City.

Because the aims of the district SLCs are to attract more citizens to participate in regular exercise (maximizing social welfare), to sustain the operations, and to generate local government revenue through utilizing operating expenditure, opening times, space, and asset, the district SLCs have initiated a series of sports activities, a number of comprehensive camps during the summer and winter vacations, and have made rental space available for diversified PA. The Department of Sports in the Taipei City Government set the unified pricing rule, an accessing fee of NT$50 per hour for gyms and NT$110 for swimming per time among twelve SLCs [30]. In addition, these district SLCs in Taipei City have begun subsidizing each senior NT$ 50 every time they use the SLCs, hoping that seniors would fully utilize the SLCs and do more exercise, leading healthier citizens [30]. Meanwhile, the department of sports set up unified opening times (6:00–22:00, seven days a week, only two days off for Lunar New Year's Eve and Lunar New Year Day of the year) for the twelve SLCs in Taipei City [30,31].

Particularly, the Department of Sports has adopted an OT management style to private firms within the contracted years; therefore, this department needs to develop a performance evaluation mechanism to assess these twelve district SLCs each year in order to ensure the citizens' safety, access rights, and generate an appropriate revenue to the local Taipei City Government. The existing evaluation and administration mechanism from the local government has adopted an expert-panel committee to assess the twelve SLCs each year. Each contracted period for each SLC is about seven years. The SLCs that perform well have the privilege to extend the next concession period. One of the advantages of the OT strategy for the local government is they can generate royalty provisions of

at least 10% of revenue for the twelve SLCs via revenue sharing, acting as a substitute for investment cost [32,33]. Meanwhile, the local government would not need to reserve an annual budget for the operating and maintenance fees for these SLCs. At the same time, they could offer citizens appropriate indoor facilities for PA.

In this context, this study discusses how an evolutionary theory can serve as an informational foundation on how the efficiency-based performance model (EPM) under OT project finance for the public district SLCs contributes to the public administration.

## 3. Literature Review

### 3.1. The Theoretical Background Evolutionary Theory

The sustainability of competitive advantage in evolutionary theory is similar to the concept of sustained competitive advantage [13]. Under Nelson and Winter's [14] framework, there are different routines to operate business in a firm. Some of these efficient and effective routines in a firm operate sustained competitive advantage in the face of competition. The difference in the company's performance across time comes mainly from their unique resources and capabilities. The evolutionary theory exploits the internal routines of the companies to develop its competitive advantages [15]. Researchers emphasized that a firm transforms the core routines to performance [34]. Another resource-based view theory proposes that each company has a set of resources and competences, and some capabilities lead to better impact on performance than the others [34]. Nelson and Winter [14] pointed out that firms differ in the routines they have established to operate their business. Some of these routines are exposed to be more efficient and effective than others. The least efficient and effective routines are either improved, or it is possible a firm will not survive in the long run. The most efficient and effective routines generate competitive advantages or superior performance for firms.

Such differences in impact are attributed to the efficiency with which a firm is able to convert its resources into valuable and difficult-to-imitate capabilities and into financial performance [35]. Efficiency is defined as the ratio of a set of outputs to that of their inputs and is measured in terms of the maximum outputs that can be attained with a given set of inputs [35,36].

### 3.2. Performance Assessment for Sport and Leisure Centres

The literature has used two traditional methods to evaluate a firm's performance. The first is ratio analysis, such as employee productivity and sales volume [34], but its applications have been limited due to the possibility that different ratios produce different performance results. The second is the parametric method, such as a regression model. This technique requires good fit and does not produce any benchmarks [36,37]. For example, Gomes and Yasin [38] used multiple regressions to measure performance in Portuguese services. Charnes, Cooper, and Rhodes (CCR) [39] developed the CCR-DEA model with the assumption of constant return-to-scale (CRS), using multiple indicators to evaluate the relative efficiency of each decision-making unit (DMU). Compared to the limitations of these two traditional methods, DEA taking all the indicators into consideration is suitable for analyzing a service organization's performance [40].

DEA has the benefit of being able to simultaneously integrate a wide range of inputs and outputs [41]. DEA is a methodology for assessing the efficiency frontier given a set of inputs being converted into a set of outputs in a production process. This estimation is non-parametric and does not require any specific assumption about the functional form of the production function generating outputs from inputs. DEA can account for several inputs and several outputs in the estimation including financial variables or survey data [36]. The DEA methodology is illustrated in the management science literature as appropriate for taking into account idiosyncratic characteristics of a decision-making unit (DMU) [41]. DEA can categorize benchmark models and establish precise guidelines for the less well-performing DMUs for further improvement, which indicates the DEA method demonstrates two very important steps of benchmarking [42]. Sherman and Zhu [43] used

the term "balanced benchmarking" to refer to DEA, which is a model for efficiency evaluation and benchmarking where multiple performance indicators are assessed in the service industry.

There are limited studies to explore the performance evaluation for sport organizations. Terrien and Andreff [44] evaluated the organizational efficiency of 36 national football leagues in Europe. They indicated that benchmarking best practices may help the less well-performing leagues to follow the best example path toward organizational improvement. O'Boyle and Hassan [45] used secondary data stemming from the Sport Development Report for Equestrian Sports 2013 to compare the organizational performance of for-profit and non-profit sport organizations. They utilized different financial indicators, such as revenue, expense, and profit, as one of the dependent variables to examine the determinants through 22 regression models. There are different determinants for different financial indicators.

Liu et al. [28] utilized the output-oriented DEA, including operating expenditure, operating hours, and space as three inputs, and revenue and number of participants as two outputs, to assess the operating performances of sport facilities and swimming pools in the UK's National Benchmark Service database. Kung and Taylor [5] also used operational data of sports facilities from the UK's National Benchmarking Service, indicating empirical results in which customer satisfaction is higher for SLCs managed by the government; however, the financial performance of government-owned SLCs is lower compared to SLCs managed by private companies.

Most studies have used the survey method to analyze the antecedents of customer satisfaction toward SLCs. Howat et al. [9] applied the survey method to investigate the links between customer satisfaction and service quality in 30 Australian SLCs. Alexandris et al. [8] investigated the determinants of customer satisfaction and service quality of private gyms. Macintosh and Doherty [46] studied the service environment and facilities in private gyms. Rafoss and Troelsen [47] indicated that the density of sports facilities is relatively high in Norway and Denmark because of government sponsorship. They further pointed out that sport activities are more popular if the government supports the sport facilities.

Therefore, this paper used mixed-method designs, which also included a modified Delphi method based on theory, literature review, and in-depth interviews with a panel of 20 experts, to achieve input/output selection consensus and derived three inputs (asset value, operating expenditure, and square meter) and two outputs (revenue and number of participants) of the quantitative EPM. This study further utilized total-factor framework [27] and formulated the quadrant-based matrix method to identify the sustainable and improvement clusters of SLCs.

*3.3. The Application of Total-factor Framework*

The studies developed a total-factor framework by using DEA [28,48], which delivered a valuable alternative to the conventional aggregated efficiency. Although information regarding aggregate efficiency is valuable, aggregate efficiency scores cannot determine which of these resources need improvement. Furthermore, a more in-depth analysis requires disaggregated data for energy efficiency across countries [27]. Most studies used the input-oriented DEA method to estimate the total-factor energy efficiency in different countries, such as Japan [49] and China [28,50]. However, there is a paucity of research to estimate the disaggregate data on the other research themes and relevant output targets. Hu et al. [51] firstly attempted to disaggregate the total-factor output efficiencies to assess the performance of ASEAN (Including Brunei, Cambodia, Indonesia, Laos, Malaysia, Myanmar, the Philippines, Singapore, Thailand, and Vietnam) Airports. Their work formulated the disaggregate output efficiency by comparing the actual output with the target output. The total surplus adjustments in output are regarded as the inefficient portion of actual output generation.

## 4. Materials and Methods

The proposed methodology mixed with the qualitative and quantitative methods is illustrated in a hierarchical framework in Figure 2.

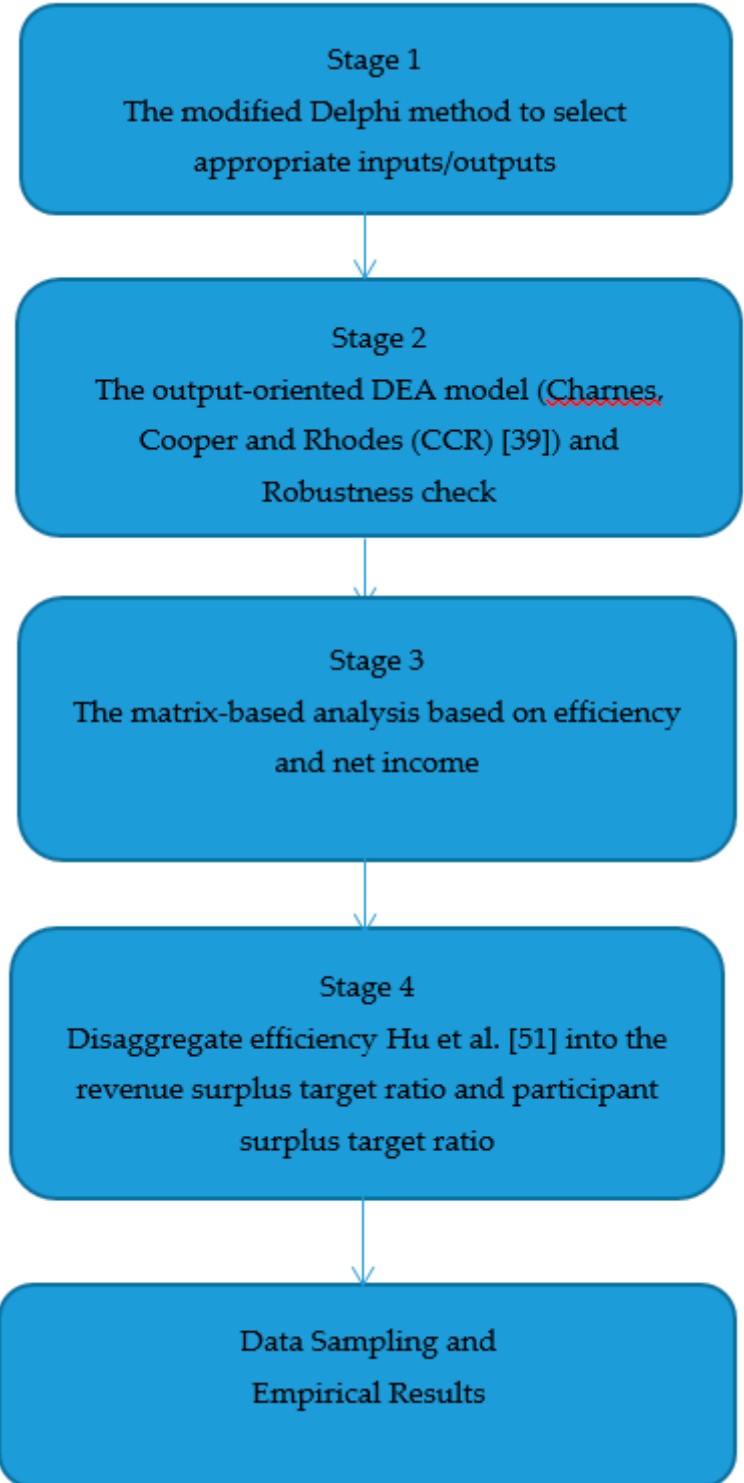

**Figure 2.** Proposed methodology.

The first stage of Figure 2 is to select the appropriate inputs and outputs using the modified Delphi method, which included the evolutionary theory, literature review, and in-depth interview from a panel of twenty sport experts. The second stage established the EPM for assessing the performance through the constant return-to-scale CCR-DEA [28,39]. The third stage is to develop the quadrant-based matrix using efficiency and net-income as two axes to identify the benchmark SLCs and less well-performing SLCs that need improvement for efficiency and net-income. Last but not the least, the stages disaggregated OSTR into the revenue target ratio and participant attraction

target ratio from the total-factor framework [51] as the improvement direction for better performance.

### 4.1. The Modified Delphi Method

First, this study used the qualitative method (a modified Delphi method) to interview twenty sports experts who operate sport facilities using the OT scheme. The interview outline is based on multiple factors retrieved from the literature to select and confirm the key inputs and outputs of the efficiency-based performance model. A two-step modified Delphi method was used to establish consensus. Twenty experts representing thirteen island-wide CEOs of SLCs in Taiwan, two senior managers from headquarter of sports group, two academic professors specialized in the sport facility, and three government officers in sports were invited to participate as the expert panel. In round 1, based on the evolutionary theory and literature [5,29], operating expenditure, operating time, space, asset, revenue, and number of participants are candidates of this model. The first-round input/output selection statements were distributed to the panel. Panel members were asked to mark "agree" or "disagree" to select these inputs and outputs beside each statement, and provide comments. The same voting method was again used for round 2. The panel experts achieved 90% consensus to select operating expenditure, space, and assets as inputs as well as revenue and number of participants as outputs of this model. The panel experts agreed to exclude the same operating hours as an input in this model because of the same operating hours (6:00–22:00) for all SLCs in Taipei City.

### 4.2. DEA Model

Second, this paper utilizes output-oriented CCR-DEA to develop this efficiency-based performance assessment model and quadrant-based matrix to identify the clusters of sustainable and improved SLCs. Based on the literature review and expert review, the research model in Figure 3 was established.

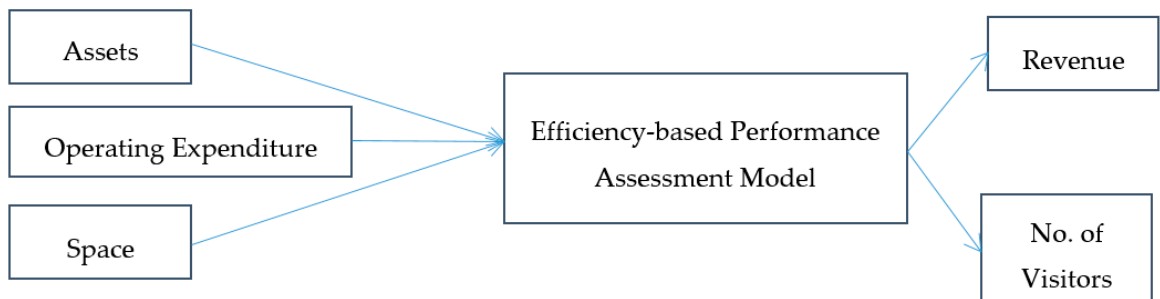

**Figure 3.** Research model**.**

Because the aims of the SLCs are to attract more citizens to participate in regular exercises, to sustain the operations, and to generate revenue, in the second stage this paper utilizes the output-oriented DEA model in Equation (1) to assess the efficiency score of each SLC as follows.

$$Max \ \phi_i$$
$$\phi_i, \lambda_1, ..., \lambda_N$$

$$s.t. \quad \phi_i y_i^m \leq \sum_{j=1}^{N} \lambda_j y_j^m, \ m = 1,...,M,$$

$$x_i^k \geq \sum_{j=1}^{N} \lambda_j x_j^k, k = 1,...,K, \tag{1}$$

$$\lambda_1,...,\lambda_N \geq 0, i = 1, 2, 3,..., N,$$

*Where* $\phi_i$ is the inverse of the efficiency score of the *i*th DMU,

$N$ is the number of SLCs,

$M$ is the number of outputs, $K$ is the number of inputs.

In the third stage, this paper sets up the four-quadrant matrix formulated by efficiency (long-term sustainability) and net income (short-term profitability) from the literature [23,26] and presents the sustainable benchmark and improvement directions.

Regarding those SLCs located in the quadrant that denotes them as less well-performing in terms of below average efficiency score, the output-surplus analysis is developed for an individual inefficient SLC based on Equation (2).

$$\Delta y_i^m = \sum_{j=1}^{N} \lambda_j y_j^m - \phi_i y_i^m, \ m = 1,...,M. \tag{2}$$

### 4.3. Estimate the Output Surplus Target Ratio (OSTR) Using the Total-Factor Framework

Technical efficiency reflects the ability of firms to generate as much output as possible with a given level of input. The summation of surplus and radial adjustments is the total amount ('target') that can be increased without increasing the input levels. With respect to revenue and number of participant outputs, the above summation is called the output surplus target (OST), and the formula is

$$\text{OST} \ (\Delta y_i^m) = \text{Non-radial Surplus Adjustment for Output + Radial Adjustment for Output.} \tag{3}$$

This paper estimates two types of OSTs, including revenue surplus target (RST) and participant surplus target (PST). An inefficient SLC can increase RST and PST without increasing input consumption.

Revenue surplus target ratio (RSTR) in the following Equation (4) can be measured based on the surplus and radial adjustments of revenue generated from the DEA model:

$$\text{RSTR} = \text{RST} \ (\Delta y_i^1) / \text{Actual Revenue Output} \ (y_i^1), \tag{4}$$

As Equation (4) indicates, the RSTR represents each SLC's inefficient level of revenue stream. Since the minimal value of RST is zero, the value of RSTR lies between zero and unity.

Participant surplus target ratio (PSTR) in the following Equation (5) can be measured based on the surplus and radial adjustments of number of participants estimated from the DEA model:

$$\text{PSTR} = \text{PST} \ (\Delta y_i^2) / \text{Actual Number of Participant} \ (y_i^2), \tag{5}$$

As Equation (5) indicates, the PSTR represents each SLC's inefficient level of number of participants.

### 4.4. Data Sampling

This paper adopts three inputs (operating expenditure, space, and asset) and two outputs (revenue and number of participant) for the 34 selected SLCs to collect the annual operational data from in-depth interviews and the official website of each SLC and Taipei City Government during three consecutive years. Meanwhile, the SLC names were anonymous due to data confidentiality. The individual sampled data were not be revealed.

## 5. Results

### 5.1. Descriptive Statistics

This paper adopted three inputs and two outputs for the 34 selected SLCs in Taipei City during three consecutive years. The main objective for the private investors of an OT project is profit or revenue, whereas the main goal for the local government is whether the operation of the OT project will give a positive social welfare to the society (how to increase number of participants in this study). Table 1 illustrates the descriptive statistics, with monetary values expressed in NT dollars, and indicates that these variables were relatively diversified. The average annual operating revenue among these SLCs in Taipei during this time approached NT$72 million. The maximum annual revenue of one SLC was nearly NT$114 million, which is quite good. The average number of users per year was close to 1 million, indicating that citizens are willing to utilize the SLC sport and leisure courses and facilities. This result is consistent with the policy aims of the sport administration of the Ministry of Education in Taiwan [52]. The operating space ranged from 6121 square meters to 66,310 square meters. The assets of SLCs also ranged from NT$8.76 million to NT$404 million. These descriptive statistics imply that traditional methods are unable to evaluate the operating performances among these SLCs due to huge operating size differences.

**Table 1.** Descriptive statistics of inputs/outputs.

| Input/output | Mean | St Dev | Max | Min |
|---|---|---|---|---|
| Assets (NT$) | 70,305,603.08 | 103,449,855.38 | 404,522,793.00 | 8,758,876.00 |
| Operating Expenses (NT$) | 67,019,457.46 | 16,402,754.61 | 97,155,200.00 | 23,029,903.00 |
| Space (square-meter) | 18,168.07 | 15,358.13 | 66,310.00 | 6121.00 |
| Revenue (NT$) | 71,958,128.00 | 20,283,514.29 | 113,833,802.00 | 19,634,285.00 |
| No. of Visitors | 915,860.00 | 200,051.33 | 1,363,118.00 | 384,819.00 |

The exchange rate was US$1 = NT$29 in October 2020.

### 5.2. Efficiency Score from the DEA Model

Based on the efficiency assessment from Equation (1), the average operating efficiency score for these 34 SLCs in Table 2 was 93.1%, indicating 7% improvement is needed to achieve the efficient frontier. Eight SLCs (24% of all observed SLCs in this case) did reach the efficient frontier, meaning that these better performing SLCs used specific internal routines to generate maximum revenue and number of participants. Table 2 further shows the efficiency and output surplus based on Equation (2), indicating each output improvement surplus value. The output surplus analysis based on Equation (4) shows that those inefficient SLCs that want to reach the efficient frontier need to increase their original revenues by 8%, or NT$5.36 million per year. Another average output surplus value was 80,172 participants or about a 9% increase in the current number of annual participants based on Equation (5). This empirical result means that the SLCs need to increase 6681 participants per month in order to hit the efficient frontier.

**Table 2.** Efficiency and output surplus for each SLC.

| No. | DMU | Score | Revenue | REV_Projection | Revenue_Surplus | RSTR (%) | No of Participant | NOP_Projection | NOP_Surplus | PSTR (%) |
|---|---|---|---|---|---|---|---|---|---|---|
| 1 | SLC1_1st_yr | 0.94 | 68,650,453 | 73,343,428 | 4,692,975 | 6.8 | 878,885 | 938,966 | 60,081 | 6.8 |
| 2 | SLC2_1st_yr | 0.85 | 19,634,285 | 23,150,847 | 3,516,562 | 17.9 | 384,819 | 453,741 | 68,922 | 17.9 |
| 3 | SLC3_1st_yr | 0.91 | 94,788,891 | 104,285,864 | 9,496,973 | 10.0 | 1,227,955 | 1,350,985 | 123,030 | 10.0 |
| 4 | SLC4_1st_yr | 0.88 | 60,499,744 | 68,964,332 | 8,464,588 | 14.0 | 773,922 | 882,202 | 108,280 | 14.0 |
| 5 | SLC5_1st_yr | 0.95 | 59,596,290 | 62,697,830 | 3,101,540 | 5.2 | 776,276 | 816,675 | 40,399 | 5.2 |
| 6 | SLC6_1st_yr | 0.96 | 85,022,564 | 89,030,527 | 4,007,963 | 4.7 | 908,487 | 1,000,083 | 91,596 | 10.1 |
| 7 | SLC7_1st_yr | 0.89 | 87,747,697 | 98,173,452 | 10,425,755 | 11.9 | 1,021,488 | 1,163,509 | 142,021 | 13.9 |
| 8 | SLC8_1st_yr | 1.00 | 61,956,210 | 61,956,210 | 0 | 0.0 | 996,361 | 996,361 | 0 | 0.0 |
| 9 | SLC9_1st_yr | 0.95 | 81,136,425 | 85,621,758 | 4,485,333 | 5.5 | 1,005,821 | 1,061,424 | 55,603 | 5.5 |
| 10 | SLC10_1st_yr | 1.00 | 113,833,802 | 113,833,802 | 0 | 0.0 | 1,298,340 | 1,298,340 | 0 | 0.0 |
| 11 | SLC11_1st_yr | 1.00 | 64,885,634 | 64,885,634 | 0 | 0.0 | 1,053,550 | 1,053,550 | 0 | 0.0 |
| 12 | SLC12_1st_yr | 0.90 | 65,644,648 | 73,109,258 | 7,464,610 | 11.4 | 736,114 | 887,867 | 151,753 | 20.6 |
| 13 | SLC1_2nd_yr | 0.96 | 72,324,693 | 75,247,968 | 2,923,275 | 4.0 | 944,480 | 982,655 | 38,175 | 4.0 |
| 14 | SLC2_2nd_yr | 0.86 | 45,050,754 | 52,403,997 | 7,353,243 | 16.3 | 751,131 | 873,732 | 122,601 | 16.3 |
| 15 | SLC3_2nd_yr | 1.00 | 62,206,105 | 62,206,105 | 0 | 0.0 | 807,004 | 807,004 | 0 | 0.0 |
| 16 | SLC4_2nd_yr | 0.92 | 59,374,530 | 64,308,039 | 4,933,509 | 8.3 | 858,959 | 930,331 | 71,372 | 8.3 |
| 17 | SLC5_2nd_yr | 0.93 | 57,522,525 | 61,710,227 | 4,187,702 | 7.3 | 788,819 | 846,246 | 57,427 | 7.3 |
| 18 | SLC6_2nd_yr | 0.99 | 81,487,201 | 82,691,671 | 1,204,470 | 1.5 | 903,156 | 944,524 | 41,368 | 4.6 |

| | | | | | | | | | |
|---|---|---|---|---|---|---|---|---|---|
| 19 | SLC7_2nd_yr | 0.89 | 85,831,468 | 95,943,234 | 10,111,766 | 11.8 | 976,590 | 1,148,279 | 171,689 | 17.6 |
| 20 | SLC8_2nd_yr | 0.93 | 59,712,861 | 64,108,947 | 4,396,086 | 7.4 | 848,838 | 911,330 | 62,492 | 7.4 |
| 21 | SLC9_2nd_yr | 0.98 | 84,033,418 | 85,545,537 | 1,512,119 | 1.8 | 961,653 | 978,957 | 17,304 | 1.8 |
| 22 | SLC10_2nd_yr | 0.99 | 112,961,762 | 113,546,085 | 584,323 | 0.5 | 1,363,118 | 1,370,169 | 7,051 | 0.5 |
| 23 | SLC11_2nd_yr | 1.00 | 68,878,589 | 68,878,589 | 0 | 0.0 | 846,530 | 846,530 | 0 | 0.0 |
| 24 | SLC12_2nd_yr | 0.91 | 74,214,523 | 81,472,345 | 7,257,822 | 9.8 | 868,344 | 1,001,975 | 133,631 | 15.4 |
| 25 | SLC1_3rd_yr | 0.98 | 79,296,794 | 80,759,183 | 1,462,389 | 1.8 | 1,003,754 | 1,022,265 | 18,511 | 1.8 |
| 26 | SLC2_3rd_yr | 1.00 | 46,018,650 | 46,018,650 | 0 | 0.0 | 1,042,918 | 1,042,918 | 0 | 0.0 |
| 27 | SLC3_3rd_yr | 0.75 | 71,463,778 | 95,562,528 | 24,098,750 | 33.7 | 826,614 | 1,105,362 | 278,748 | 33.7 |
| 28 | SLC4_3rd_yr | 0.92 | 58,837,222 | 63,762,977 | 4,925,755 | 8.4 | 853,260 | 924,694 | 71,434 | 8.4 |
| 29 | SLC6_3rd_yr | 0.79 | 69,584,894 | 87,687,215 | 18,102,321 | 26.0 | 929,058 | 1,170,750 | 241,692 | 26.0 |
| 30 | SLC7_3rd_yr | 0.74 | 66,857,793 | 90,671,942 | 23,814,149 | 35.6 | 768,160 | 1,067,986 | 299,826 | 39.0 |
| 31 | SLC9_3rd_yr | 1.00 | 87,996,020 | 87,996,020 | 0 | 0.0 | 984,126 | 984,126 | 0 | 0.0 |
| 32 | SLC10_3rd_yr | 0.97 | 121,004,391 | 124,715,882 | 3,711,491 | 3.1 | 1,274,288 | 1,441,609 | 167,321 | 13.1 |
| 33 | SLC11_3rd_yr | 1.00 | 74,119,532 | 74,119,532 | 0 | 0.0 | 877,524 | 877,524 | 0 | 0.0 |
| 34 | SLC12_3rd_yr | 0.92 | 75,773,661 | 82,080,834 | 6,307,173 | 8.3 | 1,003,593 | 1,087,129 | 83,536 | 8.3 |
| | **Average** | **0.93** | **72,880,818** | **78,249,719** | **5,368,901** | **8.03** | **927,763** | **1,007,935** | **80,172** | **9.34** |

Note: REV_Projection: Revenue projection; NOP: Number of Participants; RSTR: Revenue surplus target ratio; PSTR: Number of participants surplus target ratio; the exchange rate was US$1 = NT$29 in October 2020.

### 5.3. Robustness Validation

Sensitivity analysis with different outputs and inputs was adopted in the robustness validation of the SLCs in Table 3. First, this study developed an original model containing two outputs and three inputs to assume as the basic model (Model_0). We then excluded one input from the basic model to create three models containing two inputs and two outputs (Model_1, Model_2, and Model_3) and dropped one output from the basic model to develop two models having three inputs and one output (Model_4 and Model_5). Table 3 lists the models and all organizing inputs and outputs with this sensitivity analysis.

**Table 3.** Robustness validation and Spearman correlation.

| Variables | Original | Model_1 | Model_2 | Model_3 | Model_4 | Model_5 |
|---|---|---|---|---|---|---|
| *Inputs* | | | | | | |
| Assets (NT$) | V | | V | V | V | V |
| Operating Expenses (NT$) | V | V | | V | V | V |
| Space (square-meter) | V | V | V | | V | V |
| *Outputs* | | | | | | |
| Revenue (NT$) | V | V | V | V | | V |
| No. of Visitors | V | V | V | V | V | |
| Spearman correlation_ Original Model | 1 | 0.848** | 0.382** | 0.996** | 0.576** | 0.831** |

** This indicates *p*-value < 0.05.

This paper adopts Spearman's correlation analysis to analyze the five models. The correlation coefficients of Model_0 to Model_5 are tabulated in Table 3. All models achieved high correlation values with a significant *p*-value, suggesting that the empirical results were extremely stable.

### 5.4. The Quadrant-Based Matrix Based on Efficiency Score and Net Income

This paper utilized the four-quadrant matrix formulated by efficiency (long-term sustainability) and net income (short-term profitability) to present the sustainable benchmark and improvement directions. The analysis of four quadrants in Figure 4 based on long-term sustainability/efficiency and short-term profitability/net income provides further improvement suggestions for specific SLCs.

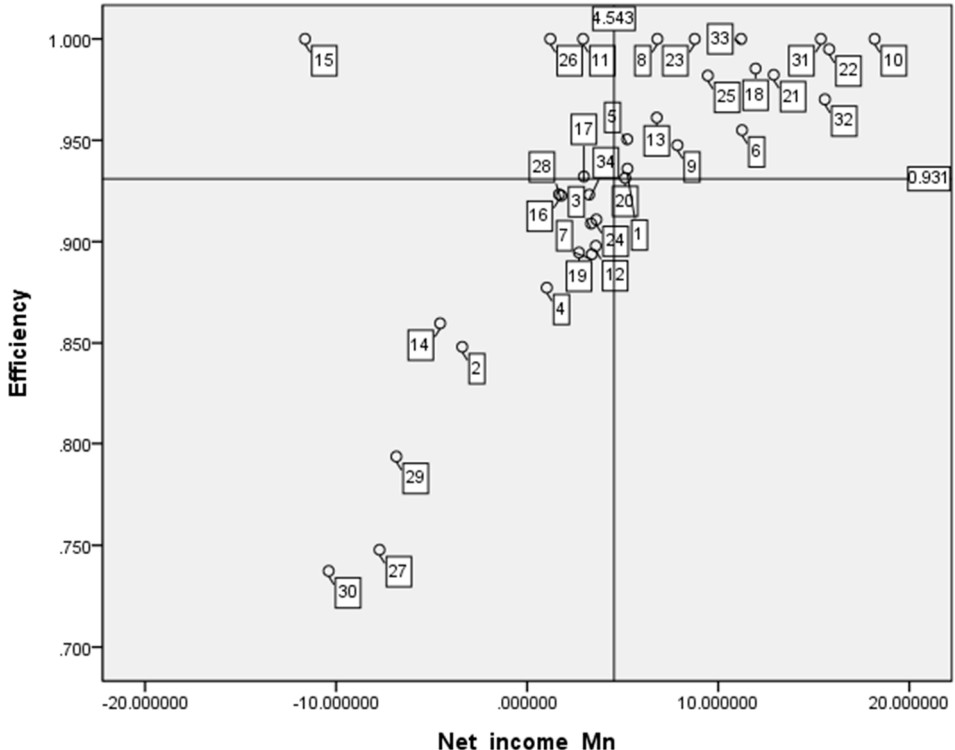

**Figure 4.** Four-quadrant analysis.

Quadrant I—This quadrant is sustainable in terms of high efficiency and high net income. This quadrant included six SLCs in the first year, five SLCs in the second year, and four SLCs in the third year, showing that the number of sustainable SLCs was on a decreasing trend. This result provides a warning message for the Taipei City Government and the management teams of OT-SLCs in the city. The SLCs located in this quadrant represent the benchmark for other less well-performing SLCs. For example, #SLC9 in the third year used relatively less operating space and less operating expenses (NT$72.6 million) to generate N$88 million in revenue and attracted almost 1 million participants, leading to it being the best sustainable performer (efficiency score = 1). This SLC utilized its internal routines, was not limited the fewer spaces, and launched more exercise courses to yield this revenue stream. Thus, it can be the benchmark for other inefficient and unprofitable SLCs.

Quadrant III—Change in operating style (low efficiency and low net income). There were five SLCs in the first year, four SLCs in the second year, and five SLCs in the third year located in this quadrant. This result reveals that approximately five SLCs need to change their operating strategy and raise their utilization rate in order to enhance efficiency and net income. This quadrant further provides an early warning alert for the Taipei City Government to enhance the frequency of auditing and to prepare for the next bidding opportunity for other interested private firms.

Quadrant II—"Improvement in Net Income" in terms of high efficiency but low net income. There was one SLC in the first year, two in the second year, one in the third year located in this quadrant. This result indicates that few SLCs here achieved high managerial efficiency, and it is necessary for them to examine their revenue and cost. However, if the cutting point of net income is zero, then only one SLC would be located in this quadrant, which means other SLCs, even classified into low net income, already achieved the break-even point of revenue and operating cost and expenditure.

Quadrant IV—"Improvement in Efficiency" in terms of low efficiency and high net income. There was one SLC in the third year located in this quadrant. This SLC needs to maximize two outputs, based on the output surplus analysis in Table 2, in order to increase managerial efficiency.

According to Table 2, for example, #SLC12_3rd_yr needs to increase NT$6.3 million in terms of revenue and have 83,536 participants so as to achieve the efficient frontier.

### 5.5. Total-Factor Output Efficiency

Aggregate efficiency scores by DEA cannot determine which of these routines needs to have the first priority improvement. Furthermore, more in-depth analysis requires disaggregated data for efficiency assessment [27,28]. OSTR improvement indexes further guided the decision makers of inefficient SLCs to maximize revenue or attract more participants in the SLCs. Since the amount of total output surplus is defined as the inefficient portion of actual output value, Equation (4) could calculate the RSTR, and Equation (5) could estimate PSTR in the Table 2. As the minimum values of RST and PST are zero, indicating this SLC has the best performance, the values of RSTR and PSTR lie between 0% (best performing SLC, no need for further improvement) and 100% (less well-performing SLCs, indicating the improvement portion). According to Table 2, the average RST was NT$5,368,901 (US$1 = NT$29 as of October 2020), the average RSTR was 8%, and the maximum RST achieved NT$24,098,750 (SLC3_3rd_yr), while RSTR was 35.6%, respectively. The average PST was 80,172 participants, and the average PSTR was 9.34%, indicating that the first priority action for less well-performing SLC is to attract more participants in order to enhance its performance.

## 6. Discussion and Implications

### 6.1. Discussion

In order to maximize the operating performances of SLCs in Taipei, the Taipei City Government used the method of OT operations to contract out SLCs to private firms for concession periods, which is consistent with that of England's public sport facilities [29]. According to the empirical results herein, the observed SLCs in Taipei achieved a 0.93 efficiency score, which is higher than that of England's public sport facilities (average efficiency score of 0.64). This result validates that the OT management strategy for SLCs in Taipei is successful. Even though there is 7% improvement for the less well-performing SLCs in Taipei City, the other city governments in Taiwan have followed this similar type of OT management strategy to contract out their SLCs to private firms. This empirical result also validates that the relatively successful strategy could be the better benchmark model for other new market players. This finding is consistent with the work from Lo and Fang [53]. Their study identified the benchmark model that is taken as the best practice for developing marketing strategy on Facebook.

Based on the empirical results from the quadrant-based matrix, there were six SLCs in the first year, five SLCs in the second year, and four SLCs in the third year located in Quadrant I (sustainable and profitable), showing that the number of sustainable SLCs is on a decreasing trend. The SLC#11 located in the Xinyi district in the third year used relatively less operating assets (NT$52,880,869 in contrast to average asset value NT$66,919,993 in Taipei City), operating space (6171 m² in contrast to average space 18,859m² in Taipei City), and less operating expenses (NT$72.6 million vs. average value NT$68.6 million) to generate N$88 million in revenue and attracted almost 1 million participants, leading to it being the best sustainable performer (efficiency score = 1). This SLC continually utilized its internal routines, did not limit spaces, and launched more profitable exercise courses with certain participants to yield this revenue stream. Thus, it can be the best practice for other inefficient and unprofitable SLCs. Table 2 empirically validated that this SLC was sustainable through achieving the efficient frontier for three consecutive years.

There were five SLCs in the first year, four SLCs in the second year, and five SLCs in the third year located in Quadrant III (low efficiency and low net income). This result reveals that approximately five SLCs need to change their operating strategy and raise their utilization rate in order to enhance efficiency and net income. This quadrant further provides an early warning alert for the Taipei City Government to enhance the frequency of auditing and to prepare for the next bidding opportunity for other interested private firms. For example, SLC#3 located in the Zhongzheng district produced NT$3.34 million and then lost NT-$11.6 million in the second year

and NT-$7.72 million in the third year. This warning information provided the committee members of Taipei City Government to terminate the OT contract of the original private company and then changed to another private firm to keep on operation for another concession period. This is the advantage of the OT project strategy.

The SLC #12 in the third year was located in Quadrant IV: "Improvement on Efficiency". According to Table 2, #SLC12_3rd_yr needs to increase revenue by NT$6.3 million and participants to 83,536 so as to achieve the efficient frontier. According to Table 2, the RST of SLC12_3rd_yr was NT$6.3 million (US$1 = NT$29 as of October 2020), and the average RSTR was 8.3%; the PST was 83,536 participants and the PSTR was 8.3%. This result indicated that this SLC#12 located in Songshan district needs to launch popular and profitable fitness courses to attract more participants and, therefore, enhance its sustainability. SLC#12 could also benchmark the best practice of SLC#11, launching a course through a combination of swimming and yoga, to be aqua-yoga courses, in order to receive relatively greater course revenue. Furthermore, the Total body Resistance exercise (TRX) suspension training course is another example to benefit the participants' total body health by offering simple equipment, effective workouts, and education.

Liu et al. [29] assessed the operating efficiency of England's public sport facilities, indicating their average efficiency score is above 0.6, and nine SLCs achieved the efficient frontier. This present paper also assesses the operating efficiency and net income of each SLC in Taipei City, showing only eight SLCs were at the efficient frontier. Hence, how to maximize SLC utilization rate is needed in order to identify their benchmark/weak areas [5] and further increase their long-term sustainability. Based on the stereotype of extant facilities, the swimming pool is a revenue-generating facility, but while it attracts more participants, it provides lower net income. Particularly, a swimming pool needs a certain level of operating expenditure to maintain water quality, undergo regular hygiene checks, and has regular audits from the Department of Health [54] and Sports of Taipei City Government [55], including the residual chlorine content being between 0.3 and 0.7 parts per million (PPM); the pH value of water quality ranging from 6.5 to 8; the total number of colonies being less than 500 colony forming units (CFUs) per 1 metric (mL) of pool water; and the *Escherichia coli* population being lower than (including) 6 CFUs per 100 metric (mL) of water [54,55]. Therefore, a swimming pool facility cannot make a profit if the SLCs merely receive the ticket fee (NT$110) from participants. The benchmark SLC (for example, #SLC11 in 3rd_yr) launched diversified swimming courses, including a combination of swimming and yoga, to be aqua-yoga courses, in order to receive relatively greater course revenue. Al Rabadi [56] suggested that offering aqua-yoga exercises has a significantly positive effect on coordination in terms of breathing and the relaxing acts for beginning swimmers. Hence, this exercise can be recognized as a new training methodology to effectively accelerate the learning process of swimming [56]. Kantyka et al. [57] further claimed that water-based exercises have many essential physical, mental, and social advantages and proved that water exercises, including aqua fat burner, aqua senior, and aqua yoga, are mostly recommended to obese, middle-aged, or elderly people. The SLC#3 was located in a relatively older community. The water-based exercise courses might probably motivate the middle-aged and elderly customers to join in this course.

Because of the relatively low ticket price (NT$50, which is equivalent to US$1.70 for a one-hour exercise) to access the fitness center, the SLCs need to launch more fitness courses that can generate more value proposition to customers so as to yield more revenue. Particularly for a low-utilization time period, like during the day, another useful alternative suggestion for less well-performing SLCs is to establish a strategic alliance with schools and enterprises in order to increase the number of participants during these low periods. The SLCs can probably develop leisure activities such as painting competition, lion dance, and traditional Taiwanese leisure activities to attract diversified customers. Wakefield and Barnes [58] indicated that casino resorts often stage extra theme parties to attract more customers during slow seasons. The SLC#12 located in Songshan district could develop a strategic alliance with the near elementary and high schools to utilize its sports venue during the day time and launch art appreciation courses to encourage leisure customers to experience this SLC. Gómez-Baya et al. [59] suggested that regular physical activities were related to lower depressive

symptoms for adolescents aged between 12 and 15 years old. One advantage for the SLC that establishes the solid relationship with the secondary school is to increase the utilization rate of the sport facilities, and another social welfare is to benefit the adolescents' mental health. The SLC would not only increase the economic performance for itself, but it would also establish sustainability for the social welfare.

This empirical result also sheds light on the importance of the evolutionary theory in the competitive advantage for SLCs, which is an area that has been under-investigated in the sports industry. This paper briefly elaborates upon the theoretical and managerial implications in the following two sections.

*6.2. Theoretical Implications*

From the evolutionary theory in the competitive advantage perspective, this paper contributes to the performance assessment model development for SLCs from the local government perspectives in several ways.

First, this paper considers multiple factors, including three input resources and two outcomes, simultaneously, that reflect the performance assessment model for SLCs. This study uses an input–output framework that organizations may investigate inside the firm to see where there are opportunities for improvement based on internal routines and help to identify gaps between expected performance and accomplished performance [15]. This efficiency-based performance model (EPM) with robustness validation could contribute to the evolutionary theory in the competitive advantage that the firm may utilize existing routines to exploit external opportunities. Superior routines are given a major role in assisting firms to achieve higher performance and sustainability.

Second, this paper takes into account a short-term financial indicator (one-year net income) together with a long-term sustainability indicator (efficiency score) as two axes to develop a four-quadrant analysis in order to identify the benchmark quadrant (high efficiency and high net income), "Improvement for Net Income" quadrant (high efficiency but low net income), "Improvement for Efficiency" quadrant (high net income but low efficiency), and "Change operating style" quadrant (low efficiency and low net income).

Particularly, the Taipei City Government contracts out the SLCs to private organizations over a certain period of time. Due to the OT operating type, the SLCs in Taiwan have both the operating goals of revenue generation and participant growth. Utilizing this matrix analysis could assist in the scientific decision making of the Taipei City Government. Finally, this paper set up the output-oriented DEA model based on the evolutionary theory in the competitive advantage due to the fact that the aims of SLCs are to increase the exercise participation rate for residents and help city governments enhance their financial structure. This model is consistent with the work of Liu et al. [29], who also used the output-oriented DEA model to assess the operating performance of public sport facilities in England. This paper enhances such a model to better the output surplus analysis so as to offer specific improvement directions for less well-performing SLCs. At the same time, it further advances the matrix analysis, which can be valuable for identifying sustainable and profitable SLCs, as it contributes to the development of the evolutionary theory in the competitive advantage. The robustness validation improves the stability of this EPM model.

The empirical results indicate that the average operating efficiency of SLCs in Taipei was 0.93 during these three consecutive periods. Output surplus analysis shows that the less well-performing SLCs need to benchmark efficient SLCs and increase NT\$5.36 million and 80,172 participants, on average, per year to achieve the efficient frontier. This finding is also consistent with the original policy and expectations of the Taipei City Government [55]. Meanwhile, taking the low-utilization periods to implement the promotional campaign in order to maximize the revenue is also consistent with revenue management in other service industries [60]. Barney [61] proposed that the firm could analyze the internal resources by using value, rareness, imitability, and organization (VRIO), which are four questions that ask if resources are valuable, unusual, costly to replicate, and not

interchangeable. Wernerfelt [62] indicated that the unique and idiosyncratic firm's resources and capabilities can lead to sustained superior performance.

Last but not the least, the OSTR from the total-factor framework indicated that increasing more participants in the SLCs would be the first priority to enhance the SLC's performance. McDougall and Levesque [63] proposed that providing the core service should be the most important. Hence, the SLCs need to identify the key value proposition in order to deliver the unique selling points to the customer segments [64]. Osterwalder and Pigneur [64] developed the business model canvas to help the decision makers identify the value proposition and the customer segments. It is also consistent with the finding of this work that attracting more target customers through the core value delivery is more important than revenue generation.

*6.3. Managerial Implications*

The research findings suggest that less well-performing SLCs need to increase their revenue stream to fulfil private business objectives and increase number of participants to fulfil the social welfare objective from the government perspective. Existing studies have suggested that offering extra attractive activities during the slow season can be a possible strategy to increase the number of customers and revenue stream [58]. McDougall and Levesque [63] recommended that managers should focus on providing value to customers, possibly through a focus on the relational aspects of service quality. Hence, less well-performing SLCs need to analyze customer surveys and observe the number of customers in each time period, such as employing volume analysis during morning, afternoon, and evening sessions. Based on such analysis, an SLC could offer relatively low-price exercise courses to increase the number of customers without any negative impact on service quality.

Utilizing marketing campaigns or media advertisements to attract more customers during slow periods is also another alternative to increase revenue. Verma and Chandra [65] suggested that sponsorship and media programs have a positive impact to draw customers. More sport events held in the SLCs would further attract more participants to visit sport venue and join in the sports courses in that SLC. Hsu et al. [66] asserted that local residents are arranged to provide volunteer services to create an atmosphere for the local sport event, and directly interact with spectators, athletes, and other event stakeholders. The sport event, therefore, would have a positive impact on the quality of life of the residents and further achieve sustainable development goals in rural communities [66].

Researchers also claimed that financial incentive strategies (such as price discounts for low periods, coupons for frequent participants, etc.) could give consumers real benefits so as to establish regular usage habits [67]. The SLCs could initiate lucky draws, promotional coupons for first-time participants, revenue management, such as low-price course in the low seasons, for loyal customers to achieve higher efficiency.

Advertising is an effective communication tool for customers. One study claimed that social media is changing buying behaviors by initiating diverse and immediate communication among potential customers and between them and organizations [68]. In this newly emerging communication environment, shared information and experiences become greater essential factors to influence customer decision-making, perhaps more important than other traditional media [69].

The less well-performing SLCs could also provide a shuttle bus between their branch and public transportation stops in order to solve distance obstacles for local residents in remote areas of Taipei. Hoekman et al. [70] indicated short walking distances to sports centers is an incentive for residents to visit sports centers in Holland. Based on a survey in 2009, the Sports Administration of the Ministry of Education [52] also supported that if the time spent going to a sports center is under 10 min, then this is acceptable by local residents.

**7. Conclusions, Limitations, and Future Research**

This study focused on the problem of performance management of OT project finance for public sports and leisure centers from the perspective of local government. The existing research on this type of OT project for public sports facilities is limited because most are focused on customer

service issues. Meanwhile, most studies tried to reduce local public expenditure by improving local services instead of simultaneously increasing social welfare and local government revenue. This paper contributes to the use of an evolutionary theory of competitive advantage to develop the EPM with robustness validation for 34 SLCs in Taipei City. The average efficiency score for these 34 SLCs was 93.1%, indicating that 7% improvement for these OT merchants would be necessary. The empirical findings suggest that less well-performing SLCs need to increase average revenue by 8.3%, or NT\$5.36 million per year and attract 9.34% (80,172) more participants, based on output surplus analysis, in order to reach the efficient frontier. Quadrant-based matrix analysis could help local governments identify the sustainable and profitable SLCs and less well-performing SLCs based on the long-term sustainability and short-term profitability. The empirical results indicated there were 15 SLCs located in the sustainable benchmark quadrant with high sustainability and high profitability, albeit with a decreasing trend. The quadrant with low sustainability and low profitability included 14 SLCs, showing that the local government should consider changing to another OT merchant in the next concession period. There were four SLCs located in Quadrant II, indicating the profitability index should be improved, and there was one SLC located in Quadrant IV, which requires efficiency improvement.

This study provides policy makers of the local government a scientific reference to keep or drop the current operating private enterprise in the next concession period. The total-factor framework disaggregating the efficiency into an innovative OSTR provides local government with a contracted period to manage the SLCs through further specific improvement advice. The most underperforming SLCs could follow this proposed quadrant analysis and OSTR index, utilizing their internal resources to develop more attractive and reasonably priced exercise courses for participant growth.

The main contribution of this study is its development of a new model for evaluating the sustainable and profitable performance of SLCs, although the research is subject to several limitations. First, because none of the SLCs in Taiwan is a publicly listed company, it is a challenge for researchers to obtain operating data; thus, our selective data would limit the generalizability of these findings to other local government management. Future research could increase the observed SLCs to include more SLCs outside of Taipei City as long as SLCs in other cities have operated for more than one year. Meanwhile, future research including in-depth interviews for SLC managers and policy makers would identify more information and contribute to local government management.

Second, this paper only used financial and operating data to assess sustainability and profitability for SLCs. Future research could further expand the research objectives to include the energy expenses as one of the inputs of this model to develop a more comprehensive sustainability index.

Third, this study selected three years as the observed period for 34 SLCs. Future studies could include more research periods to examine the relationship between the sustainability index and the profitability index in a longitudinal study. Meanwhile, cultural differences research in SLCs between Asia and Western countries would be another future study. Further studies regarding to the sport tourism would also create another revenue stream for the local government in rural areas. Roman et al. [71] recommended that the use of tourism attractiveness to build competitive advantage and attract tourists would benefit the local community. Future research might examine the impact of sport events in conjunction with rural tourism on the environment and social benefits.

**Funding:** Taiwan's Ministry of Science and Technology (MOST 105-2410-H-003-048; MOST 106-2410-H-003 -107 -MY2) and the National Taiwan Normal University, Taiwan.

**Conflicts of Interest:** The authors declare no conflicts of interest.

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
