# Peer review of "Perspective of Local Government on the Performance Assessment of District Sports and Leisure Centers"

_sustainability, doi:10.3390/su12219094_

Round 1
Reviewer 1 Report
The author of the manuscript presented the perspective of local government on the evaluation of the performance of sports and entertainment. In the manuscript the author presented a very important point, although the manuscript has some drawbacks that should be corrected before publication. The topic presented in the manuscript is interesting.
The last paragraph of the introduction: "Section 7 concludes and indicates the future research". This is not true because the last chapter is 6.
Stylistic, editorial errors.
What are the directions for the future, e.g. for in-depth research, analysis? (I propose to write in the conclusions, there are only 2 short sentences about it, very general).
The manuscript was sent to Sustainability. There are only 3 short sentences about it. This needs to be expanded.
I propose to expand single sentences (lines 14-15, 109-111 and others).
What's new in this article? Why is the subject matter so important and new?
The conclusions are very short (only 149 words, 1 short paragraph).
There is discussion in the manuscript, but no critical discussion.
The subsection numbering is incorrect (4.2-5.2, 4.3-..., 4.4-..., 4.5-...). Page 10, 13, 15.
Citations: [4,8,9,10,11] can be replaced with [4,8-11] etc.
Why was the DEA model chosen? I propose to describe in more detail.
I propose to expand the literature (e.g. in applications) with publications on sports, tourism, recreation, etc.
- Gómez-Baya, D.; Calmeiro, L.; Gaspar, T.; Marques, A.; Loureiro, N.; Peralta, M.; Mendoza, R.; Gaspar de Matos, M. Longitudinal Association between Sport Participation and Depressive Symptoms after a Two-Year Follow-Up in Mid-Adolescence. Int. J. Environ. Res. Public Health 2020, 17, 7469.
- Roman, M.; Roman, M.; NiedzióÅ‚ka, A. Spatial Diversity of Tourism in the Countries of the European Union. Sustainability 2020, 12, 2713.
- Hsu, B. .-Y.; Wu, Y.-F.; Chen, H.-W.; Cheung, M.-L. How Sport Tourism Event Image Fit Enhances Residents’ Perceptions of Place Image and Their Quality of Life. Sustainability 2020, 12, 8227.
DOI is missing in a few articles (reference list).
Author Response
As the attachment.

Reviewer 2 Report
Page 7: Please explain what the modification of the Delphi method used in this paper is.
Pages 7-9: As none of the methods used are entirely new, please provide bibliographic references indicating their source.
Page 9: Please clarify which DMUs are analyzed through the DEA. The paper refers to both SLCs and sectors.
Also, It should also be specified whether the technical efficiency score took into account CCR or VRS assumptions.
Author Response
as the attachment.

Round 2
Reviewer 1 Report
I accept the manuscript in the current version. Good luck!